# Estimated Incidence and Prevalence of Serious Fungal Infections in Morocco

**DOI:** 10.3390/jof8040414

**Published:** 2022-04-17

**Authors:** Badre Eddine Lmimouni, Christophe Hennequin, Richard O. S. Penney, David W. Denning

**Affiliations:** 1Parasitology and Medical Mycology Laboratory, Military Hospital Teaching Mohammed the Fifth, BioInova Research Center, Faculty of Medicine and Pharmacy, University Mohammed the Fifth, Rabat 10100, Morocco; b.lmimouni@um5r.ac.ma; 2Service de Parasitologie-Mycologie, Hôpital Saint-Antoine, AP-HP, 75012 Paris, France; christophe.hennequin-sat@aphp.fr; 3Centre de Recherche Saint-Antoine, CRSA, Inserm, Sorbonne Université, 75012 Paris, France; 4Global Action for Fungal Infections, 1208 Geneva, Switzerland; rpenney@gaffi.org; 5Manchester Academic Health Science Centre, Faculty of Biology, Medicine and Health, University of Manchester, Manchester M13 9PL, UK

**Keywords:** *Aspergillus*, *Candida*, *Cryptococcus*, *Pneumocystis*, keratitis

## Abstract

Few data are published from Morocco on fungal disease, although numerous case reports attest to a wide range of conditions in the country. Here, we estimate for the first time the incidence and prevalence of serious fungal diseases in the country. Detailed literature searches in English and French were conducted for all serious fungal infections. Demographic and individual underlying condition prevalence or annual incidence were obtained from UNAIDS (HIV), WHO (TB) and other international sources. Deterministic modelling was then applied to estimate fungal disease burden. Morocco’s population in 2021 was 36,561,800. Multiple publications describe various fungal diseases, but epidemiological studies are rare. The most frequent serious fungal infections were tinea capitis (7258/100,000) and recurrent vulvovaginal candidiasis (2794/100,000 females). Chronic pulmonary aspergillosis is also common at a prevalence of 19,290 (53/100,000) because of the relatively high rate of tuberculosis. The prevalence of asthma in adults exceeds one million, of whom fungal asthma (including allergic bronchopulmonary aspergillosis (ABPA)) probably affects 42,150 (115/100,000). Data are scant on candidaemia (estimated at 5/100,000), invasive aspergillosis (estimated at 4.1/100,000), HIV-related complications such as cryptococcal meningitis and *Pneumocystis* pneumonia and mucormycosis. Fungal keratitis is estimated at 14/100,000). Mycetoma and chromoblastomycosis are probably rare. Fungal disease is probably common in Morocco and diagnostic capacity is good in the teaching hospitals. These estimates need confirmation with methodologically robust epidemiological studies.

## 1. Introduction

Serious fungal infections have generally been increasing in frequency, but a lack of diagnoses limits the usefulness of some epidemiological surveys. For example, the diagnosis of invasive aspergillosis is often missed [1], and if the WHO essential diagnostic tests *Aspergillus* antibody, *Histoplasma* antigen or *Pneumocystis* PCR are not routinely available, then chronic pulmonary aspergillosis, disseminated histoplasmosis and *Pneumocystis* pneumonia will be grossly under-diagnosed. The common lack of diagnosis relegates much fungal disease to clinical oddities and curiosities, while the actual toll that these infections take is under-appreciated. Here, we attempt to show that, in Morocco as well as in many low- or middle-income countries, many, but not all, fungal diseases are relatively common and need pro-active diagnosis to be appropriately managed in the clinic.

Morocco is a predominantly Muslim country in North Africa with a population of nearly 37 million (11th in Africa) [2]. Split into French and Spanish protectorates in the first half of the 20th century, it has been an independent monarchy since 1956. Morocco’s economy is the 5th-largest in Africa by GDP and 13th in terms of GDP per capita (2020) [3].

There are very few data on the burden of fungal diseases in Morocco, unlike many of the Arab League countries [4]. We have estimated the incidence and prevalence of the most serious fungal diseases using national, regional and international data in specific populations at risk. Our attempt to collate what is known and published provides a national gap analysis, which can be addressed.

## 2. Methods

This literature review was based on articles about fungal infections using the Google Scholar and PubMed/Medline search engines, African newspapers, health reports, epidemiological journals in Morocco and WHO reports. The articles searched were in English and French. The keywords searched were: fungal infection, opportunistic disease, HIV/AIDS, tuberculosis, chronic pulmonary, cryptococcosis and histoplasma, all associated with Morocco. There are few published studies on fungal diseases in Morocco. Where no article was found from Morocco or North Africa, we used data from other countries outside the continent.

The socio-demographic data were taken from the *CIA World Factbook* [2]. HIV prevalence and AIDS deaths were taken from the 2019 UNAIDS report [5]. We assumed that HIV-infected people not on antiretroviral therapy (ART) develop profound immunodeficiency over 7 years (77% of cases are B subtype [6]) and that the rate of failure of ART (measured primarily by ART resistance) is 11%. Pulmonary tuberculosis (TB) annual incidence and mortality was taken from the WHO global report 2020 [7]. Asthma prevalence in adults was taken from a survey of 10,051 interviewees in 14,289 households conducted in 2008 [8]. Chronic obstructive pulmonary disease (COPD) prevalence was derived from the BREATHE study with an estimated 10.5% admitted to hospital annually, as in Algeria [9,10]. Data on lung cancer were taken from the Global Cancer Observatory [11] and those on acute myeloid leukaemia (AML) from the WHO expert committee on middle-income countries [12]. Only renal and autologous bone marrow transplants are performed in Morocco [13]. Liver transplantation has been performed since 2019 (2 at the Mohammed V military teaching hospital in Rabat and 8 at the Rabat University hospital.

The various populations and denominators were used to estimate prevalence or incidence of different fungal diseases, as described in other papers [14]. The assumptions used to compute annual incidence and prevalence are also shown.

## 3. Results

Morocco’s population in 2021 was 36,561,800, of whom 27% were children 14 years old or younger: 9,614,960. There are an estimated 10,256,400 women between the ages of 15 and 54 years. In 2020, UNAIDS estimated that 22,000 were living with HIV, of whom about 16,500 were undertaking ART. There were about 36,000 new cases of tuberculosis in 2020, of which about 18,000 were pulmonary and only 410 had a co-infection with HIV. The survival rate of TB is about 93%. Asthma prevalence in adults and children was documented in 2008 at 3.89% of the population, or just over 1 million adults. The COPD prevalence (GOLD stage 2–4) was estimated as part of the international BREATHE study, and 2.2% of the population was documented with it—a total of 775,980 people, of whom about 84,500 are admitted to hospital each year. There are 7530 lung cancer cases annually, and about 800 AML cases. Only about 40 renal and 5 allogeneic haematopoietic stem cell transplants are carried out annually. 

Overall, our estimate is that approximately 3,300,000 people in Morocco (9% of the population) suffer from a serious fungal infection. This total is heavily dominated by tinea capitis in children, which is probably common, but perhaps not as common as our estimate (Figure 1).

### 3.1. Pulmonary Fungal Diseases

Invasive aspergillosis (IA) was assumed to complicate 13% of AML patients and an equal number of all other haematological malignancies, lymphoma and multiple myeloma, at an estimated total of 210 cases of IA annually [15]. In addition, an estimated 2.6% of lung cancer patients (n = 7353 in 2020) develop IA [16], a total of 190 patients. Among the 85,450 people with COPD admitted to hospital, 1100 (1.3%) probably develop IA [17,18]. Overall, the annual incidence of IA is about 1500 patients (4.1/100,000). 

Chronic pulmonary aspergillosis (CPA) may be mistaken for pulmonary TB, be a co-infection during or in the weeks after completion of antituberculous therapy, or may develop in the years after taking the TB cure, especially in those with residual pulmonary cavities. In 2020, 18,360 pulmonary TB cases were reported in Morocco, of whom 8% were not proven, and overall, 3056 patients died. Many patients with aspergilloma and CPA are described in Morocco, usually in surgical series [19,20,21,22,23]. The estimated annual and post-TB prevalence is shown in Table 1. This assumes a 19% rate of CPA in undocumented TB cases [24] and a 10% rate of CPA at the end of antituberculous therapy and in the 6 months after this [25,26], and a 6.5% and 0.2% annual rate in those with and without cavitation at the end of antituberculous therapy (22%) [27,28]. It also assumes a 20% rate of first-year mortality of CPA and 7.5% thereafter [29,30]. Overall, the annual incidence of CPA related to TB is estimated at 2482 cases with 165 deaths in the immediate 12 months after first presentation with possible TB. The 5-year period prevalence is estimated at 11,551 with an additional 791 deaths annually. Assuming that TB is the underlying pulmonary condition in 60% of the patients [31], a total CPA prevalence of approximately 19,290 (53/100,000). *Aspergillus fumigatus* would appear to be the most common pathogen in CPA in Morocco [22].

The percentage of adults with asthma in 2008 was estimated at 3.89%, which means about 1,038,000 are affected. Of these, an estimated 26,000 have allergic bronchopulmonary aspergillosis (ABPA) (2.5%) [32]. Assuming that 10% of these asthmatic adults have poorly controlled and severe asthma, and that 33% of these people are sensitised to fungi, we estimate that 34,260 have severe asthma with fungal sensitisation (SAFS) [33]. There may be some duplication between these groups, and if this is 30%, then about 42,200 adults have ‘fungal asthma’ in Morocco (115/100,000). ABPA was described many years ago in Morocco [34].

A few distinctive cases of fungal rhinosinusitis have been described in Morocco [35,36]. It is not possible to estimate the burden, but, if it is as common as in Israel, then 5% of the population may be affected by allergic fungal rhinosinusitis [37].

### 3.2. HIV-Related Fungal Diseases

Amongst Morocco’s population of HIV patients, an estimated 2600 in 2020 were at risk of a serious optimistic infection. While *Pneumocystis* pneumonia has been reviewed as a topic in Morocco [38], data on its incidence are lacking. Data were published some years ago for Tunisia [39]. Assuming an incidence of 15% (a general figure for many countries [40]), the annual incidence in HIV patients is likely to be 195 patients. Cryptococcal meningitis is described in Morocco [41,42,43,44,45]. Cryptococcal meningitis is less frequent, occurring in 2.9% of advanced HIV disease cases (the general figure used for the eastern Mediterranean countries) [46] so an estimated 160 patients are likely affected annually. Oesophageal candidiasis is a common problem in HIV patients, and assuming that 20% of those with low CD4 counts and 5% of those on ART are affected, it is likely that 1350 patients are affected at least annually. There are probably a small number of cases of histoplasmosis in Morocco [47,48,49], but it is not possible to estimate the burden.

### 3.3. Invasive and Superficial Candidiasis

Although candidaemia is described in Morocco, few studies collating its incidence are published, and none with a general population denominator. In children with leukaemia, *Candida* spp. accounted for 14% of healthcare-associated infections (invasive aspergillosis was not diagnosed) [50]. The risk factors in intensive care include implanted catheters and broad-spectrum antibacterial agents, as in other countries [51]. *Candida albicans*, *C. glabrata* and *C. tropicalis* were the most frequent pathogens. We have assumed a conservative annual incidence of 5/100,000, which converts to about 1800 cases [52,53]. As in other countries, about 33% of cases occur in intensive care, including neonatal and burn units. Using data derived from France, where peritoneal (intraabdominal) candidiasis is 50% as common as candidaemia in ICUs [54], we anticipate 275 cases annually (Table 1).

Oral and vaginal candidiasis are common problems, but not generally too serious. However, we estimated the more problematic recurrent vulvovaginal candidiasis in pre-menopausal women at a 6% rate [55]. This computes to over 510,000 women affected in any given year. One cross-sectional study of 114 consecutive women referred to gynaecology specialists found 22.8% had a positive microscopy for *Candida* spp. [56]. *Candida albicans* was isolated most frequently (69.2%), followed by *Candida glabrata* and *Candida tropicalis* (15.5% each). The most commonly affected age group was 25–35 years. The literature is silent on recurrent vulvovaginal candidiasis for Morocco.

### 3.4. Skin and Eye Infections

Tinea capitis is reported by several series in Morocco [57,58,59,60,61,62]. A recent meta-analysis of tinea capitis in Africa estimated that 23% of African school-aged children, or 2,663,557 (95% CI 1,968,716–3,358,398) and 7285/100,000, are affected in Morocco. This could be an over-estimation, as rates of tinea capitis may be influenced upwards by countries in sub-Saharan Africa.

Mycetoma has been reported infrequently in Morocco over many years [63,64,65,66,67,68,69], and the country is above the ‘Mycetoma’ belt [70]. About 50% of the cases are fungal, or eumycetoma [70]. To date, 18 cases of chromoblastomycosis have been reported in Morocco [71]. No cases of sporotrichosis have been reported in Morocco. 

Mucormycosis is described in Morocco, but is probably uncommon or rare [72,73,74,75]. One case of cutaneous mucormycosis in an immunocompetent child has been reported [72], as well as one case of rhinofacial mucormycosis [75].

Fungal keratitis is a serious and often blinding condition usually related to a minor eye injury or wearing of contact lenses. While occasionally reported in Morocco [76,77], no large series have been published. We have therefore used the data from Egypt to estimate annual incidence—14/100,000, or 5120 cases.

## 4. Discussion

In Morocco, life expectancy at birth in women is 74 years and in men, 72 years [78], and 27% of the population is under the age of 15 [2]. There is no compulsory or universal health insurance scheme. In 2007, 16% of the population had some form of medical insurance, including 11% of the population, which was covered by public-sector insurers [79]. Most of the population identify as Arabs or Arab-Berbers and are Sunni Muslims. Very few people from sub-Saharan Africa live in Morocco, despite the geographical proximity.

Diagnostic provision in Morocco for fungal diseases is reasonable in teaching hospitals. There are nine university hospital centres linked to a school of medicine, and in these hospitals, there are eight parasitology and mycology departments and 16 specialists on the topic: in Rabat, Casablanca, Marrakesh, Agadir, Fez, Meknes, Oujda and Tangier. In Laayoune, a university hospital is being developed. The lack of epidemiology publications on large populations or cohorts of patients reflects a lack of time—clinical and teaching commitments leave little time for surveillance or research. 

Diagnostic capacity is good in the teaching centres, with all offering microscopy, culture and histopathology and almost all having cryptococcal and *Aspergillus* and *Candida* antigens, as well as *Aspergillus* antibodies. *Pneumocystis* PCR is only available in Casablanca and Rabat, and *Histoplasma* antigen testing is not available. It is not clear if the country has cases of histoplasmosis in humans (as opposed to the relatively frequent equine histoplasmosis (epizootic lymphangitis) caused by a separate subspecies, *H. farciminosum*). Opportunities exist in strengthening their capacity with external funds from, for example, the Global Fund for AIDS, TB and Malaria [80].

The major limitation of this work is the lack of large studies from the country. Almost all the estimates are inferences. For example, we found only one paper on vulvovaginal candidiasis and none on recurrent disease, yet it is unlikely that Moroccan women are not immune to this troublesome complaint. There are considerable bodies of work on chronic pulmonary aspergillosis and aspergilloma, principally surgical series [19,20,21,22,23], with a note that this disorder is relatively frequent. There are multiple publications focused on HIV infection, but fewer on the complications of AIDS [38,41,42,43,44,45]. Several Moroccan authors have summarised the clinical approach to particular conditions, such as *Pneumocystis* pneumonia or chronic pulmonary aspergillosis, indicative of local awareness of these conditions.

The estimation approach to chronic pulmonary aspergillosis differs from prior estimates, which have primarily focused on post-tuberculous CPA. Here, we have included estimates of misdiagnosed TB (which is probably a numerically small problem in Morocco as 94% of cases are confirmed bacteriologically) and cases occurring during and immediately after therapy. Again, all the assumptions made are based on studies from other countries, and given the genetic components of CPA susceptibility, these estimates are likely to be inaccurate. They require local studies to be carried out. 

The very large numbers of children with tinea capitis may well be an over-estimation. In the one study from Morocco, 18% of children attending hospital had tinea capitis, but this is clearly a select population. 

This survey supports previous reports from other North African countries that about 9% of Moroccan inhabitants suffer from fungal infections. The most important, as perceived by their incidence/prevalence, are pulmonary fungal infections (mostly *Aspergillus* diseases), both superficial (tinea capitis) and mucosal (vulvo-vaginal candidiasis). These population-based estimates should promote the implementation of a fungal surveillance system to describe more precisely the landscape of these infections, which should drive public healthcare policy.

## 5. Conclusions 

The teaching hospitals and clinical leaders are well-acquainted with serious fungal disease in Morocco, but there is clearly a lack of diagnostic capacity and awareness in smaller hospitals and a major lack of high-quality epidemiology research on this topic. The development of several lateral flow assays for fungal antigens and antibodies has opened up the realistic possibility of every local hospital and specialised HIV or TB clinics being able to screen. Likewise, training materials (in French) for direct microscopy for fungi and histopathology (www.microfungi.net, accessed on 20 February 2022) should be used to facilitate rapid detection of some fungal diseases. The listing of several key diagnostics by the WHO should encourage the Ministry of Health to ensure complete country coverage of Morocco’s citizens for fungal disease, combined with laboratory and clinical training. Wide diagnostic implementation could be supplemented with surveillance to track incidence and trends over time. Overall, there is an immediate opportunity to significantly improve clinical outcomes and probably reduce unnecessary empirical therapy with both antibacterial and antifungal therapy and directly contribute to the control of antimicrobial resistance. 

## Figures and Tables

**Figure 1 jof-08-00414-f001:**
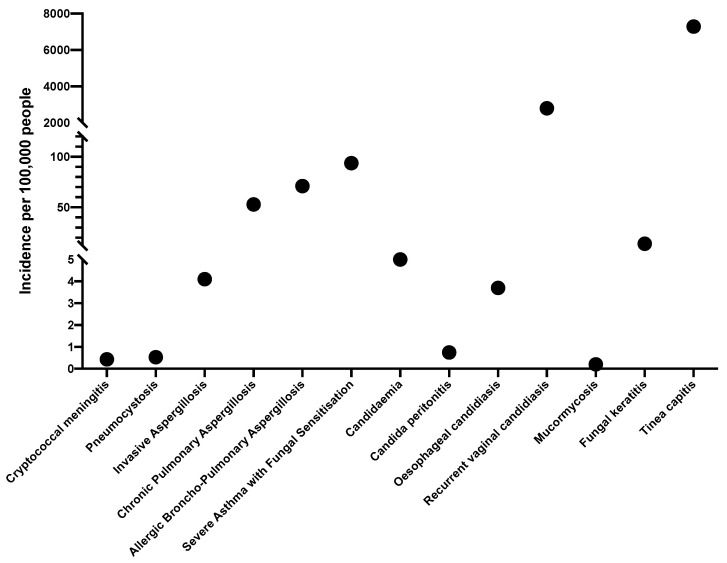
The estimated population annual incidence and prevalence of selected invasive fungal diseases in Morocco.

**Table 1 jof-08-00414-t001:** The estimated annual caseload (incidence or prevalence) of serious fungal infections in Morocco and number per 100,000 people.

Fungal Infection	Predominant Groups at Risk	Rate Per 100,000	Estimated Number of Cases
Cryptococcal meningitis	AIDS	0.43	160
PCP	AIDS	0.53	195
IA	Haematological malignancy, lung cancer and 1.3% of COPD admissions to hospital	4.1	1500
CPA	Tuberculosis patients and other respiratory disorders	52.8	19,290
ABPA *	Adult asthma patients	71.0	25,950
SAFS *	Adult asthma patients	93.7	34,260
Candidaemia	Hospitalised patients	5.00	1830
Candida peritonitis	Post-surgical patients	0.75	275
Oesophageal candidiasis	HIV infection	3.7	1346
Recurrent vaginal candidiasis ^#^	Adult women	2794	510,740
Mucormycosis	Multiple, especially diabetes	0.20	73
Fungal keratitis	Corneal injury, contact lens	14.0	5120
Tinea capitis	4–14-year-old children	7285	2,664,000
**Total burden estimated**			**3,305,100**

PCP, pneumocystis pneumonia; IA, invasive aspergillosis; CPA, chronic pulmonary aspergillosis; ABPA, allergic bronchopulmonary aspergillosis; SAFS, severe asthma with fungal sensitisation. * Duplication between ABPA and SAFS is likely as both are sensitised to *Aspergillus*. The total number of fungal asthma cases is around 42,150 (115.3/100,000). ^#^ rate per 100,000 females only.

## Data Availability

All applicable data are published and referenced in the paper.

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
