# Peer review of "Estimated Incidence and Prevalence of Serious Fungal Infections in Morocco"

_jof, 2022, doi:10.3390/jof8040414_

Round 1

Reviewer 1 Report

Dear Author,

  1. See the comments in attached pdf file. 

Author Response

Editorial Review 1

Write the abstract as per journal format. Rewrite in sentence form without any subheadings.

Done

L56 better to define the year chosen for the study

Response: We have added the year the GDP etc was estimated.

L60 Add one table of your term used for the search and no. of disease reported after filter.

Response: This is not possible because very relevant articles were found and often those that were relevant were identified not on PPUBMED/MEDLINE, but in our files and from reference lists of other articles.

L63 Where no article was found from Morocco or North Africa, we used data from other countries outside the continent" then why mentioned in title "...serious fungal infections in Morocco".

Response: Because the estimation if for Morocco and these other papers were used for the estimation.

Results As aspergillosis and other pulmonary diseases are important part of study no causal organisms and their management effort by WHO or local government mention.

Response: We have added info on the predominant causative organism in CPA. The lack of government action is typical, and this paper should enable such action to be taken. Until we did this work, it was not known that aspergillosis was such a problem.

L182 what is the % of incidence

Response: Mucormycosis is probably rare. Table 1 gives the estimated incidence (2 per million).

L235 add conclusion from the study. what is the future prospect of this study. how it support Moroccan populations.

Response: A concluding paragraph has been added.

Reviewer 2 Report

The authors intended to make a revuew on fungal infection in Morocco, filling a ga in the present literature.

However, as the authors state, the conclusions are based mainly on inferential data and hypotheses, which lowers the scientific soundness of the paper.

The data are presented in a fragmentary manner, so that the reading becomes difficult.

The English needs to be greatly revised, some sentences for example, lack the verbs.

The paper must be extensively revised to gain scientific value.

Author Response

Review 2

The authors intended to make a review on fungal infection in Morocco, filling a gap in the present literature.

However, as the authors state, the conclusions are based mainly on inferential data and hypotheses, which lowers the scientific soundness of the paper.

The data are presented in a fragmentary manner, so that the reading becomes difficult.

The English needs to be greatly revised, some sentences for example, lack the verbs.

The paper must be extensively revised to gain scientific value.

Response: The english has been reviewed and improved again.

Reviewer 3 Report

Do you intend to write similar article about other countries? It can be good idea to compare incidence and prevalence of serius fungal infections in different countries from Africa.

Author Response

Review 3

Do you intend to write similar article about other countries? It can be good idea to compare incidence and prevalence of serious fungal infections in different countries from Africa.

Response: Many country burdens are published, including about 20 from Africa. See: https://gaffi.org/media/country-fungal-disease-burdens/  Work is currently ongoing for Mali, Eritrea and Tunisia, and it is anticipated that these will be published in 2022.

Reviewer 4 Report

Dears authors

1- Discussion:
to deepen in consideration of the problem of antifungal resistance the use of drugs used in Morocco and natural methods. 

2 - Check the bibliographic entries throughout the text, some of which are non-compliant.

3- review the tables going into the specifics of the species paying attention to the statistical analysis

4 - Review English grammar and in particular applied scientific English: in particular verb tenses and syntax in the discussion.

Author Response

Review 4

1- Discussion:
to deepen in consideration of the problem of antifungal resistance the use of drugs used in Morocco and natural methods. 2 - Check the bibliographic entries throughout the text, some of which are non-compliant.

3- review the tables going into the specifics of the species paying attention to the statistical analysis.

Response: The causative species for candidemia and VVC are given in the text. A paper suggests that A. fumigatus is the predominant species in CPA and we have added this (L130).

It is not clear what additional “statistical analysis” is required?

4 - Review English grammar and in particular applied scientific English: in particular verb tenses and syntax in the discussion.

Response: We have carefully reviewed the English.

Reviewer 5 Report

On line 197, substitute "hosptals" for "hospitals".

--> Why did the authors used data from Egypt to estimate annual incidence in Marocco? What do these countries have in common in terms of fungal infections epidemiology?

--> The incidences (even if estimated) could be graphed to improve the reader's view of the data.

Author Response

Review 5

On line 197, substitute "hosptals" for "hospitals".

Response: This has been corrected

--> Why did the authors used data from Egypt to estimate annual incidence in Morocco? What do these countries have in common in terms of fungal infections epidemiology?

Response: Unfortunately, there is a lack of many specifically Moroccan studies focused on fungal infections. This is one of the reasons ofr such papers to promote the initiation of surveys that would depict the landscape of fungal infections that would help clinicians, biologists and public health professionals responsible to pay attention and develop conditions for improving the management of these infections in their country. So in the case of the lack of local study we used regional data. While Egypt strictly speaking does not belong to the Maghreb, some aspects (climate, culture, economic situation…) resemble Morocco.

--> The incidences (even if estimated) could be graphed to improve the reader's view of the data.

Response: We have added a figure showing the incidence and prevalence graphically. Thank you for the suggestion.

Round 2

Reviewer 1 Report

Please improve conclusion section and mention the future prospect of your work. 

Author Response

The conclusion has been strengthened and now reads like this.

The teaching hospitals and clinical leaders are well acquainted with serious fungal disease in Morocco, but there is clearly a lack of some diagnostic capacity and awareness in smaller hospitals and a major lack of high quality epidemiology in this topic. The development of several lateral flow assays for fungal antigen and antibody has opened up the realistic possibility of every local hospital and specialised HIV or TB clinic being able to screen. Likewise training materials (in French) for direct microscopy for fungi and histopathology [www.microfungi.net], should facilitate rapid detection of some fungal diseases. The listing of several key diagnostics by the WHO should encourage the Ministry of Health to ensure complete country coverage of Morocco’s citizens for fungal disease, combined with laboratory and clinical training. Wide diagnostic implementation could be supplemented with surveillance to track incidence and trends over time. Overall, there is an immediate opportunity to significantly improve clinical outcomes and probably reduce unnecessary empirical therapy with both antibacterial and antifungal therapy and directly contribute to the control of antimicrobial resistance.   

Reviewer 2 Report

The paper has been revised and improved in some sections. I still find the paper not easy to read, however, globally it sounds better compared to the previous version.

Author Response

Further small improvements to the english made.